# Central and Peripheral Inflammation: A Common Factor Causing Addictive and Neurological Disorders and Aging-Related Pathologies

**DOI:** 10.3390/ijms241210083

**Published:** 2023-06-13

**Authors:** Angélica P. Escobar, Christian Bonansco, Gonzalo Cruz, Alexies Dagnino-Subiabre, Marco Fuenzalida, Ignacio Negrón, Ramón Sotomayor-Zárate, Jonathan Martínez-Pinto, Gonzalo Jorquera

**Affiliations:** 1Centro de Neurobiología y Fisiopatología Integrativa (CENFI), Instituto de Fisiología, Facultad de Ciencias, Universidad de Valparaíso, Valparaíso 2360102, Chile; angelica.escobar@uv.cl (A.P.E.); christian.bonansco@uv.cl (C.B.); gonzalo.cruz@uv.cl (G.C.); marco.fuenzalida@uv.cl (M.F.); ignacio.negron@uv.cl (I.N.); ramon.sotomayor@uv.cl (R.S.-Z.); 2Stark Neurosciences Research Institute, Indiana University School of Medicine, Indianapolis, IN 46202, USA; 3Department of Anatomy, Cell Biology and Physiology, Indiana University School of Medicine, Indianapolis, IN 46202, USA; 4Instituto de Nutrición y Tecnología de los Alimentos (INTA), Universidad de Chile, Santiago 7830490, Chile

**Keywords:** neuroinflammation, neurodegenerative diseases, addictive behavior, epilepsy, anxiety, inflammaging

## Abstract

Many diseases and degenerative processes affecting the nervous system and peripheral organs trigger the activation of inflammatory cascades. Inflammation can be triggered by different environmental conditions or risk factors, including drug and food addiction, stress, and aging, among others. Several pieces of evidence show that the modern lifestyle and, more recently, the confinement associated with the COVID-19 pandemic have contributed to increasing the incidence of addictive and neuropsychiatric disorders, plus cardiometabolic diseases. Here, we gather evidence on how some of these risk factors are implicated in activating central and peripheral inflammation contributing to some neuropathologies and behaviors associated with poor health. We discuss the current understanding of the cellular and molecular mechanisms involved in the generation of inflammation and how these processes occur in different cells and tissues to promote ill health and diseases. Concomitantly, we discuss how some pathology-associated and addictive behaviors contribute to worsening these inflammation mechanisms, leading to a vicious cycle that promotes disease progression. Finally, we list some drugs targeting inflammation-related pathways that may have beneficial effects on the pathological processes associated with addictive, mental, and cardiometabolic illnesses.

## 1. Introduction: Cellular and Molecular Mechanisms of Brain and Peripheral Inflammation

Inflammation is a process characterized by the activation of immune and non-immune cells that protects the host from pathogens or damage induced by different injuries. This process eliminates the injury and promotes tissue repair and recovery [1]. In this term, metabolic and neuroendocrine responses are triggered to maintain metabolic energy and the activated immune system [1]. The inflammatory environment induces the activation of T cells, CD8+ cytotoxic T lymphocytes (CTLs), and CD4+ helper T cells (Th), triggering their proliferation and differentiation into effector T cells [2]. The production of cytokines such as interferon-gamma (IFN-γ) and interleukin (IL)-17 by the Th cells contributes to the proinflammatory state, activating macrophages and B cells, leading to systemic inflammation [3]. In addition, the autonomic nervous system innervates primary and secondary lymphoid tissues, sites where lymphocytes are produced and functioning, respectively [3]. The sympathetic nervous system, which releases noradrenaline, exerts local and systemic control of immune cells through β2 adrenergic receptor activation, which inhibits pro-inflammatory cytokine synthesis, regulating the response of T and B cells [4].

The inflammatory process is resolved when the tissue is repaired or cicatrized, named acute inflammation. However, in some diseases, the inflammatory process could be mild but persistent in time, causing chronic low-grade inflammation, which involves the participation of immune cells but also parenchymal cells in the periphery and the brain [1]. The environment and the nutritional status are the cornerstones of the development of chronic low-grade inflammation in which an imbalance occurs in favor of proinflammatory cytokines over anti-inflammatory ones [5]. Sedentarism, high-calorie diets, obesity, gut microbiota disturbances, and chronic stress have been proposed as causes promoting this imbalance [5]. Intriguingly, this chronic low-grade inflammation is a substrate for cardiometabolic and neurologic or neuropsychiatric disorders as well [5,6,7].

Emerging evidence shows that the central nervous system (CNS) and the immune system establish strong bidirectional crosstalk [8,9], which is essential to regulate both synaptic transmission and synaptic integrity [10,11]. This crosstalk is exerted through the blood–brain barrier, which is a physical and metabolic barrier present in the microvasculature of the CNS that can maintain the CNS homeostasis controlling the transport of nutrients and metabolites and immune cell trafficking. The integrity of this barrier and the recruitment of immune cells into the brain parenchyma are modulated by different chemokines, pro-inflammatory cytokines, and reactive oxygen species (ROS) produced by glial cells that are an active part of the immune response in the CNS [12,13,14,15]. It has been observed that peripherally activated T cells can enter the CNS in the absence of neuroinflammation, being the only peripheric cells that can reach the CNS in healthy conditions. This entrance allows them to reach perivascular or subarachnoid spaces, where they can interact with tissue-resident antigen-presenting cells [16]. In many diseases, such as multiple sclerosis, Alzheimer’s disease and Parkinson’s disease, the integrity of the blood–brain barrier is altered, allowing the infiltration of plasma proteins such as fibrinogen and IgG as well as immune cells such as B-cells, neutrophils, monocytes, dendritic cells, and T-cells into the CNS parenchyma, leading to deleterious effects on CNS function and thus clinical signs of disease [14,15,17]. In the mammalian brain, microglia and astrocytes are the main glial cells. Microglia are CNS-resident mononuclear phagocytes that monitor the functional state of the surrounding environment through cell ramifications that contact neurons, astrocytes, and blood vessels [18]. Depending on the insult grade, microglia extend their ramifications to surround and phagocytize the damaged area or change their morphology and release proinflammatory mediators [18]. Astrocytes are heterogenous glial cells involved in maintaining CNS homeostasis; some of the astrocyte-mediated functions are delivery of energy to neurons by the astrocyte–neuron lactate shuttle, uptake of glutamate, controlling its extracellular levels, and modulation of calcium variations that affect neuronal activity [19]. Moreover, astrocytes also mediate the immune response in the CNS, which in part depends on the crosstalk with microglia [12]. In addition, glial cells are intimately involved in the active control of neuronal activity, synapse development, synaptic neurotransmission, and plasticity [20,21]. Astrocytes and microglia release cytokines, chemokines, or thrombospondins and directly regulate the morphology and integrity of central synapses. Recent evidence highlights the main role of cytokines in synaptic transmission and plasticity. IL-1β, IL-2, IL-6, IL-8, IL-18, IFN-α, IFN-γ, and tumor necrosis factor (TNF)-α regulate the long-term plasticity and induce changes in the hippocampal learning and memory tasks [22,23]. Moreover, the microglia participate in synaptic pruning, which occurs through the interaction between the dendritic spines and glia. Indeed, synaptic pruning depends on P2Y receptor activation by purinergic molecules released by microglia [24]. Moreover, in the developing hippocampus, the postsynaptic density protein (PSD)-95 was found inside microglial cells, suggesting that microglia directly participate in the synaptic pruning process [25]. Functionally, the glia-released TNF-α is critical to synaptic scaling, a homeostatic synaptic process necessary for the proper function of central synapses. TNF-α, via an increase in the expression of β3 integrins, induces the accumulation of AMPA receptors at the synapses [26]. Together, the data show that microglia and inflammatory molecules have an important role in the structural, functional, and synaptic plasticity of neurons.

Besides the physiological role of glial cells and cytokines in neuronal synaptic function and development, they also participate in neuroinflammation leading to synaptic dysfunction and neurological and psychiatric diseases [27,28]. Neuroinflammation is a complex and well-orchestrated response of neural tissue to trauma, infection, or diseases and is mediated by glial cells that release inflammatory mediators and produce reactive oxygen and nitrogen species [29]. These inflammatory mediators impact astrocyte function, since astrocytes express receptors for cytokines and chemokines, such as transforming growth factor (TGF)-β receptor, glycoprotein (gp)130, IFN-γ receptor, and IL-17 receptor, among others, responding to signals coming from the microglia and macrophages [30]. Additionally, astrocytes, through the regulation of the blood–brain barrier, regulate the trafficking of peripheral immune cells to the brain [31,32]. During ischemia, oxidative stress, or trauma, astrocytes become phagocytic and antigen-presenting cells due to a local increase in inflammatory cytokines such as IFN-γ [33,34]. Further, astrocytes respond to chronic injuries by altering the expression of many genes, along with morphological and functional changes, a process called reactive astrogliosis [35]. During this process, the glial fibrillary acid protein (GFAP), the main constituent of intermediate filaments in astrocytes, is upregulated and used as a biomarker of reactive astrogliosis [35]. In the rodent brain, the activation of the gp130/activator of transcription 3 (STAT3) signaling pathway through cytokines, such as TGF-α, ciliary neurotrophic factor (CNTF), IL-6, leukemia inhibitory factor (LIF), and oncostatin M, directly triggers reactive astrogliosis. Alternatively, this process can be triggered indirectly through the effects of these cytokines on microglia, neurons, or endothelial cells [35,36]. Together, the data show that in the brain, microglia and astrocytes, as well as the release of proinflammatory cytokines, might induce chronic inflammation.

In the periphery, low-grade inflammation has been associated with several chronic conditions such as metabolic syndrome, nonalcoholic fatty liver disease (NAFLD), type 2 diabetes, cardiovascular disease, and aging-related pathologies [37]. As an example, in the liver, the development of NAFLD is characterized by a pathological crosstalk between the adipose tissue and the liver, where chronic inflammation is a common key [38]. The levels of proinflammatory proteins such as IL-6 in the adipose tissue of obese patients could be 100 times higher than in healthy individuals; meanwhile, high mobility group box 1 (HMGB1) protein can reach levels as high as twice those observed in control subjects. Both signals induce secretion of inflammatory cytokines by macrophages, and HMGB1 is considered an inflammation alarmin [39]. Even HMGB1 may potentiate the secretion of IL-6 and other cytokines through the binding to the receptor for advanced glycation end products (RAGE) [40]. M1 macrophages infiltrated in the adipose tissue appear to be the principal source of proinflammatory factors that end up being secreted into the blood [41]. These circulating inflammatory factors may spread chronic inflammation to other tissues [38]. Indeed, visceral adipose tissue transplantation from obese mice to high-cholesterol diet-fed mice produces an increase in hepatic macrophage content, worsening liver damage [42]. In control diet-fed mice, adipose tissue transplantation induced an elevation in circulating and hepatic neutrophils in animals who received fat transplantation from obese mice compared to those receiving transplantation from lean mice [42]. A liver with NAFLD presents an important amount of resident and infiltrated monocytes, macrophages, neutrophils, and innate lymphoid cells, which are involved in the development of chronic inflammation, NAFLD onset, and progression [43]. Hepatocytes and liver sinusoidal endothelial cells also can participate in the inflammatory pathophysiology of the disease [43].

Importantly, the peripheral inflammation observed in NAFLD has been correlated with CNS inflammation and damage. In a murine model of NAFLD induced by the injection of the hepatotoxic substance CCl_4_, severe neurodegeneration, pyknosis, vacuolations, and cavitations in the brain, plus alterations in activities of neurotransmitters catabolizing enzymes, were observed [44]. In mice, NAFLD induced by a high-fat diet (HFD) caused neuroinflammation with elevated numbers of activated microglial cells, increased the inflammatory cytokine profile, and increased expression of toll-like receptors in the brain, as well as neuronal apoptosis [45]. Accordingly, recent studies have found that NAFLD is associated with poorer cognition in human patients [46]. Data suggest a strong correlation between low-grade chronic inflammation in the periphery with neuroinflammatory processes.

In the next sections, we will discuss chronic inflammation observed in other peripheral tissues key in the pathophysiology of cardiometabolic and/or aging-related diseases, such as those in skeletal muscle and the heart.

## 2. Factors Triggering Inflammation: Lifestyle

### 2.1. Diet and Body Weight

Animal models with diet-induced obesity have high levels of pro-inflammatory cytokines such as TNF-α, IL-1β, and IL-6 [47,48] and activation of microglia and astrocytes in the hypothalamus [49,50,51]. The hypothalamic inflammation associated with obesity has been correlated with dysfunction in food-intake-regulating hormones such as leptin, insulin, and ghrelin [52,53,54] and with the reduction in the activity of anorexigenic POMC/CART neurons [55,56], with the consequent alterations in the regulation of food consumption and whole-body energy balance. Remarkably, just 1 day of HFD feeding induced the expression of proinflammatory biomarkers such as EGF-like module-containing mucin-like hormone receptor-like (EMR)1, ionized calcium-binding adapter molecule (IBA)1, IL-6, and TNF-*α* in the mouse hypothalamus [57]. Similar results were observed in rats, where expression of proinflammatory genes was increased after 1 day of HFD and hypothalamic gliosis was evident after 3 days, even before any sign of inflammation in the liver or adipose tissue and also prior to evident weight or fat gain [56]. This suggests that hypothalamic neuroinflammation associated with high-fat foods is one of the first steps of inflammation associated with obesity and occurs independently of peripheral inflammation.

Besides the hypothalamus, inflammation associated with obesity has also been reported in other brain areas such as the hippocampus, amygdala, cortex, and brainstem [58]. For example, elevated levels of IL-6, IL-1β, and TNF-α mRNAs and reduced expression of brain-derived neurotrophic factor (BDNF) were found in the hippocampus of mice lacking leptin receptor (db/db mice), a genetic model of obesity. Microglia and astrocytes have been proposed as the source of pro-inflammatory cytokines in the brain [59,60]. Nuclear factor kappa-light-chain-enhancer of activated B cells (NF-κB) represents a family of inducible transcription factors that can regulate genes involved in processes of the immune and inflammatory responses [61]. NF-κB signaling is important in the activation of gliosis associated with obesity; for example, acute (24 h) or chronic (6 weeks) feeding with a HFD + high sugar diet (HSD) was unable to induce hypothalamic inflammation in mice where NF-κB signaling was inhibited in GFAP-positive cells (mainly astrocytes) [50,62]. Moreover, NF-κB inhibition in astrocytes caused hypophagia, reduced weight gain, and improved glycemic homeostasis in mice chronically fed with HFD + HSD compared to control animals fed with the HFD + HSD [50], showing that glial inflammatory signaling is key for inducing whole body metabolic perturbations [59]. Like astrocytes, microglia have a capital role in the regulation of energy homeostasis. Forced microglia activation, through a cell-specific deletion of A20 (a NF-κB negative regulator), caused hypothalamic microgliosis, bone-marrow-derived microglia infiltration to the hypothalamus, increased food intake, and weight gain in mice fed with a chow diet [49]. On the contrary, depleting resident microglia in the CNS reduced food intake and diminished total body fat and weight gain in HFD-fed mice [49]. These molecular proinflammatory findings in genetic-induced obesity models were accompanied by behavioral dysfunctions, such as anxiety-like behaviors and alterations in spatial recognition memory [63]. Similar observations were reported in the hippocampus of diet-induced obesity models [64,65]. These data suggest that brain inflammation is an early step in the deleterious process induced by an unhealthy diet where the activation of NF-κB signaling in astrocytes and microglia is a key event.

Chronic low-grade Inflammation is also considered a key element in the pathophysiological pathways responsible for insulin resistance in peripheral organs of obese patients and animal models [66,67]. Indeed, target organs for insulin actions—such as adipose tissue and the liver—have been reported to suffer immune cell infiltration and pro-inflammatory cytokine production by resident and infiltrating leucocytes and parenchymal cells [66,67,68]. Similarly, as has been described for the CNS, cytokines such as IL-1β, TNF-α, and IL-6, among many others, have been associated with the inhibition of insulin signaling pathways in adipocytes and hepatocytes [68,69,70]. Elevated free fatty acids (FFA) in the blood can also act as proinflammatory factors in insulin target organs through the binding to pattern recognition receptors, such as toll-like receptor (TLR)4 in the cell membrane [71].

Skeletal muscle is also a main target organ for insulin actions, being the principal regulator of post-prandial hyperglycemia management [72]. Inflammation and elevated pro-inflammatory cytokine levels have been identified in skeletal muscle from obese individuals, linked to muscle adipose depot expansion and immune cell infiltration [73,74]. Accordingly, skeletal muscle myocytes secrete higher levels of cytokines in obese type 2 diabetes (T2D) patients [75], suggesting an intrinsic role of skeletal muscle cells in the development of obesity-related inflammation and insulin resistance. Recently, Jorquera’s group described that skeletal muscle fibers obtained from HFD-fed mice release high levels of ATP to the extracellular media, through pannexin-1 channels, activating an inflammatory response leading to insulin resistance [76]. The role of extracellular ATP and pannexin-1 channels in brain inflammation induced by diet or obesity remains unexplored.

Again, the relationship between peripheral and central inflammation has been reported, where the transplantation of visceral white adipose tissue from a diet-induced obese mouse or a db/db mouse to a lean recipient mouse caused increased hippocampal IL-1β expression, microgliosis, and long-term potentiation (LTP) deficits [77,78]; meanwhile, transplant from obese mice deficient in inflammatory activation signals (NLRP3 KO) caused no effects in the brain of recipient animals [78], indicating that the pathological crosstalk between peripheral organs and CNS structures could be mediated by these inflammatory patterns [79].

### 2.2. Stress: Effects of Glucocorticoids on Inflammation in the Context of Eustress and Distress

Stress is defined as a nonspecific biological response of an organism to any real or perceived threat from the environment that affects its homeostasis [80]. The main biological purpose of stress responses is to restore homeostasis and adapt to environmental threats or stressors [81]. Stress is positive (eustress) when animals adapt to environmental stressors [82]. When eustress occurs, the organization of stress responses is mainly regulated by two primary systems, the autonomic nervous system and the hypothalamus–pituitary–adrenal (HPA) axis [83]. Autonomic responses to stress increase heart frequency, blood pressure, and glucose availability, which in turn make energetic substrates available for stress adaptation [83]. Stressors increase the release of the corticotropin-releasing factor from the hypothalamus, inducing the release of the adrenocorticotropic hormone from the anterior pituitary, which in turn stimulates the secretion of glucocorticoids from the adrenal cortex that are going to bind to their receptors, glucocorticoid and mineralocorticoid receptors, in the limbic system and peripheral tissues [83]. The main glucocorticoid of stress responses is cortisol in humans and corticosterone in rodents [83]. To understand the etiology of stress-related mental disorders, such as major depressive disorder and anxiety disorders, the concept of stress was related to the concept of “allostasis”, defined as the adaptive process of preserving stability in response to stressful conditions [84]. When the energy cost of adaptation (allostatic load) is too high, stress induces negative health consequences [82,85]. Thus, persistent, strong, and uncontrollable stressors can lead to a maladaptive response called distress. At clinical levels, chronic distress-induced neuroinflammation is a key factor in the pathophysiology of mood disorders [86], mainly major depressive disorder [87,88]. Peripheral and central pro-inflammatory cytokines, including IL-1β and C reactive protein, are elevated in the plasma and cerebrospinal fluid in depressed patients [89,90,91]. It is proposed that the increase in neuroinflammation triggers morphological changes in limbic nuclei that regulate stress responses and emotional memories, affecting the functional connectivity between them [92,93].

The pattern of glucocorticoid secretion is circadian: when locomotor activity increases, so do glucocorticoid levels [94]. Glucocorticoids bind mineralocorticoid receptors with ten-fold higher affinity compared with glucocorticoid receptors [95,96]. Under resting levels of glucocorticoids, the mineralocorticoid receptors are mostly bound to the glucocorticoids, and the glucocorticoid receptors are not active [97,98]. Thus, when the body faces a stressor, the onset of the stress response is mediated by the mineralocorticoid receptors, which activate the HPA axis and increase the glucocorticoids [94]. Microglia have two states: one related to normal resting conditions associated with maintenance of the neural cell microenvironment, and another associated with a reactive inflammatory state (M1) [99,100,101], resulting in the release of several pro-inflammatory cytokines, such as IL-1β, IL-6, and TNF-α [101,102]. The mineralocorticoid receptors are also expressed in the CNS, where the microglia could respond to the increased levels of glucocorticoid achieved during the stress response. Mineralocorticoid receptor activation in the microglia promotes morphological changes toward the pro-inflammatory phenotype M1 [98,103,104], resulting in the release of several pro-inflammatory cytokines, such as IL-1β, IL-6 and TNFα [101,102]. Later, the activation of glucocorticoid receptors counteracts this inflammatory state, avoiding potential damage to the body that is the biological cost (allostatic load) associated with adaptation to stress [105,106]. A healthy glucocorticoid secretion pattern during the stress response allows glucocorticoid receptors to counteract the pro-inflammatory action of mineralocorticoid receptors, since when glucocorticoid receptors dimerize, they bind to motif DNA sequences called glucocorticoid-response elements, inhibiting the synthesis of several inflammatory cytokines, such as IL-1, IL-2, IL-6, IL-8, and IL-12, inducing the apoptosis of lymphocytes [107,108,109,110] and suppressing T-helper 1 cell differentiation [111]. This strong anti-inflammatory role of glucocorticoids is widely known at the clinical level, mainly in the treatment of inflammatory diseases such as allergies and rheumatoid arthritis [112] with synthetic glucocorticoids such as dexamethasone.

The role of glucocorticoids in inflammation significantly changes when the body does not adapt to stress and allostatic overload is triggered [94,113]. As a result of this condition, a pathological pattern of glucocorticoid secretion promotes neuroinflammation by both mineralocorticoid and glucocorticoid receptors [94]. Several cell types have been well-recognized to mediate the neuroinflammatory process in the brain. However, as we mentioned above, microglia and astrocytes play a key role in neuroinflammation [114,115]. When glucocorticoid levels increase compared to resting levels, mineralocorticoid receptors in the microglia are activated, inducing the proinflammatory M1 phenotype [98,103,104]. In addition, high levels of glucocorticoids also promote the activation of glucocorticoid receptors in the microglia, inducing a state of altered responsivity, which is expressed as priming the microglia to a M1 state more responsive to inflammatory signals [116,117].

Some people who are exposed to distress do not get sick or develop a mood disorder; these people are called “resilient individuals” [118]. Interestingly, preclinical studies support the idea that resilient and stress-vulnerable subjects respond differently to neuroinflammation. Social defeat stress (SDS), produced by repeatedly exposing a naive rodent to an aggressive partner [119], triggers neuroinflammation in the prefrontal cortex, amygdala, hippocampus, and nucleus accumbens (NAc) in mice vulnerable to stress [120,121]. In contrast, stress-resilient mice do not show these alterations one day after exposure to social defeat stress [114,122,123]. Remarkably, Dagnino’s group found that the resilience to stress is lost 2 weeks after SDS exposure, as shown by depression-like behaviors in previously resilient rats [124]. In addition, these rats showed neuroinflammation markers in the hippocampus, such as activated microglia (IBA-1) and astrocytes (GFAP) [124]. This suggests that neuroinflammation is involved in the pathogeny of depressive behaviors induced by stress. Therefore, suppression of neuroinflammation could be a therapeutic strategy to prevent depression-like behaviors [125].

## 3. Consequences of Inflammation Inducing Central and Peripheral Diseases

The maintenance of a bad quality of life characterized by bad nutrition, overweight, stress, chronic consumption of drugs of abuse, aging, and reduced microbiota diversity might induce the molecular, cellular, and pathophysiological processes related to inflammation, which in turn may lead to the development of several central diseases such as eating disorders, Alzheimer disease, epilepsy and anxiety disorders, as well as chronic peripherical illnesses such as sarcopenia, cardiopathy, and cardiovascular disorders. In this section, we will address inflammation as a common factor for the development of pathological conditions, affecting the quality of life and health.

### 3.1. Neuroinflammation as a Key Mediator in Addictive Behavior

#### 3.1.1. Hypercaloric Diet Induces Neuroinflammation in Brain Areas Associated with Addictive Behaviors

Chronic exposure to highly palatable diets led to an excess of weight and obesity [126]. In this context, diet-induced obesity produces alterations in the immune response that lead to a chronic inflammatory process associated with metabolic diseases such as diabetes, hypertension, dyslipidemia, and heart diseases, among others [66]. However, the inflammatory processes induced by hypercaloric diets are not restricted to the periphery but also reach brain areas that ultimately affect behavior. Indeed, patients with eating disorders present addictive behaviors that lead to urgently consuming obesogenic foods and can produce symptoms such as withdrawal syndrome when the intake of these foods is reduced [127], suggesting modifications at the level of the reward system. Indeed, a human study using the MRI technique showed that children with obesity have more gliosis and neuroinflammation in reward-associated brain areas [128]. Diet modification in animal models has revealed some of the mechanisms leading to neuroinflammation in the reward circuitry that might underlie the development of eating disorders. A recent study showed that feeding male mice for 6 weeks with a cafeteria diet increased mRNA expression of IL-1β and IFN-γ and induced microglial activation in the NAc [129]. Interestingly these modifications were reversed by treating mice with minocycline, a tetracycline antibiotic with anti-inflammatory activity due to inhibition of microglial activation. In addition, HFD increased the expression of other proinflammatory genes, such as IKKβ, GFAP, IBA-1, IL-1β, IFN-γ, and CD45 in the NAc [117]. Remarkably, the microinjection of an adenoviral construct to inhibit IKKβ directly into the NAc reversed the upregulation of these pro-inflammatory genes [130], indicating a direct effect of the HFD inducing neuroinflammation in this reward circuitry nucleus and suggesting a mechanism by which consumption of hypercaloric diets might alter behavior.

The consumption of hypercaloric diets might not only affect the inflammatory state of the individual who consumes it, but also produce transgenerational changes in behavior. Indeed, maternal overnutrition during pregnancy and the postnatal period induces in the offspring a compulsive food seeking behavior [131] and a higher preference for junk food [132]. Moreover, maternal overnutrition also promotes in the offspring a higher consumption of drugs of abuse such as alcohol, cocaine, and amphetamine [133,134], altogether indicating modifications in the offspring’s reward circuitry due to maternal overnutrition. Indeed, the administration of a cafeteria diet to pregnant rats induced in the offspring a higher sensitivity to ghrelin, a hormone promoting feeding behavior, together with microglia activation in the hypothalamus, which might explain the exacerbated feeding behavior observed in the offspring [135]. Interestingly, the exposure to a hypercaloric and highly palatable diet (Ensure) in female rats from postnatal day (PND) 23 to 65 increased astrocyte activation measured as an increased expression of GFAP in the medial prefrontal cortex (mPFC), NAc, and arcuate nucleus [136]. Conversely, male offspring (PND 50) showed decreased GFAP expression in the same brain areas, showing a transgenerational adaptive phenotype [136] and suggesting sex-dependent factors.

Altogether, this evidence suggests that neuroinflammation produced by a hypercaloric diet may induce neuroplastic changes in brain structures related to the processing of rewards that ultimately might affect the relationship between the individual and food, inducing eating disorders and drug addiction. Moreover, the evidence indicates that these modifications can be permanent and inheritable by future generations.

#### 3.1.2. Drugs of Abuse Induce Neuroinflammation in Brain Nuclei Associated with Addictive Behavior

Another important factor in the generation of inflammation, especially neuroinflammation, is the consumption of drugs of abuse. The addictive potential of drugs of abuse lies in their direct action in the reward circuitry where they act as a reward [137]. It is largely known that stress is one of the major triggers for relapse in people with substance use disorder [138]. The long periods of confinement during the COVID-19 pandemic could, therefore, have increased the consumption of drugs of abuse that have long-lasting deleterious effects on people. During the last decade, several works have studied the inflammatory effects of drugs of abuse in the brain, especially in areas involved in memory and learning, such as the prefrontal cortex (PFC) and the hippocampus. However, the neuroinflammatory effects produced by drugs of abuse in brain areas of the reward system have been little studied. In this section, we will describe the pro-inflammatory effects of the most common and accessible drugs of abuse: alcohol, psychostimulants, opioids, and tobacco, in the reward system, effects that could underlie their addictive potential. Additionally, we will parallel these data with potential pharmacological treatments that have shown benefits in reducing neuroinflammation induced by drugs of abuse.

Alcohol-induced neuroinflammation: Ethanol is undoubtedly the most widely abused drug in the world, associated with several systemic and brain effects. It has been shown that intermittent ethanol treatment for 2 weeks increases protein levels of TNF-α and IL-17A in the PFC of female adolescent mice [139]. The ethanol consumption and the up-regulation of cytokines in PFC were reduced with the injection of nalmefene, an opioid antagonist. Interestingly, the nalmefene effect was associated with an inhibition of ethanol-induced TLR4 activation [139]. On the other hand, chronic ethanol intake (9 weeks) increased protein expression of high mobility group box 1 (HMGB1) and TNF-α, and it reduced protein levels of glutamate transporter-1 (GLT-1) and metabotropic glutamate receptor-5 (mGluR5) in the NAc shells of male drinking rats [140]. In this context, HMGB1 is a danger-associated molecular pattern protein that can bind RAGE to activate NF-κB signaling, promoting the expression of inflammatory genes such as TNF-α, IL-1β, IL-6, IL-12p40 and cyclooxygenase [61]. Interestingly, ampicillin (a β-lactam antibiotic) plus sulbactam (a β-lactamase inhibitor) treatment up-regulated GLT-1 and mGluR5 in NAc shell, reducing HMGB1, RAGE and TNF-α [140], suggesting a potential therapeutic role for this antibiotic in the reversal of alcohol-induced inflammation in the reward circuitry.

Finally, alcohol consumption during pregnancy might affect the inflammatory response of the reward circuitry in the offspring. Indeed, rats exposed prenatally to ethanol (between gestational days 8 to 20) had a structural modification of the ventral tegmental area (VTA) microglia where a reduction in microglial branch and junction numbers [141] was observed. In addition, prenatal ethanol exposure induced a reduction in the protein expression of Fkbp5 (an immunophilin protein) and Cxcl16 (a mediator of innate immunity) in the NAc [142]. In this context, the expression of Cxcl16 is induced by inflammatory cytokines such as IFN-γ and TNF-α [143], suggesting a compensatory effect in the offspring to deal with inflammation.

Psychostimulant-induced neuroinflammation: Methamphetamine (N-methylamphetamine or METH) is a psychostimulant drug used to treat attention-deficit/hyperactivity disorder (ADHD) and obesity, but it is often used recreationally with a high potential to produce addiction. Regarding neuroinflammation, METH (3 mg/kg) administration in mice produced conditioned place preference (CPP) and up-regulation of CC-chemokine ligand 2 (CCL2) protein levels in PFC and NAc [144]. CCL2 is produced by microglia, neurons, activated astrocytes, and mononuclear phagocytes [145]. CCL2 binds CC-chemokine receptor 2 (CCR2) to produce inflammatory markers. In this context, the co-administration of METH with RS504393 (an antagonist of CCR2) attenuated METH-induced CPP [144], indicating that blockage of the induction of this proinflammatory mechanism reduces the rewarding properties of METH. On the other hand, METH increased TLR4-dependent NF-κB activity in microglial BV-2 cells [146], and a challenge with lipopolysaccharide (LPS) 24 h after METH administration (5 mg/kg, four times at two-hour intervals) increased NAc protein levels of IBA-1, TNF-α and IL-6 in mice [147], showing that chronic treatment with METH makes these brain nuclei more susceptible to other proinflammatory molecules. Interestingly, the administration of minocycline or SCH-23390 (an antagonist of D1 receptors) reduced NAc protein levels of IBA-1, TNF-α, and IL-6 induced by METH [147]. Remarkably, METH-triggered proinflammatory effects were observed after its acute administration. Thus, a single injection of METH (1 mg/kg) increased the mRNA expression of CD11b, TNF-α, and IL-6 in VTA, but not in PFC or NAc [146]. Moreover, acute METH administration (10 mg/kg) in rats increased protein levels of IL-1β in VTA, NAc, and PFC, which were reversed with the intra-cisterna magna administration of box A, an antagonist of HMGB1, showing that this signaling mediates part of the METH-induced neuroinflammation [148].

Other common psychostimulants, such as ecstasy (3,4-methylenedioxymethamphetamine or MDMA) and cocaine, also produce neuroinflammation. MDMA administration in adult or middle-aged mice increased GFAP-positive cells in substantia nigra pars compacta (SNpc) and striatum [149]. On the other hand, cocaine self-administration increased mRNA expression of IL-1β and GFAP in rat VTA [150]. In addition, intra-VTA cocaine (15 mg/kg) or LPS (an activator of TLR4) administration restored cocaine self-administration behavior in rats that underwent extinction training. In this context, intra-VTA administration of LPS-RS (an antagonist of TLR4) reduces cocaine-primed reinstatement [150], revealing that VTA inflammation is crucial to induce compulsive cocaine-seeking behavior.

Opioid-induced neuroinflammation: Opioids include natural molecules from opium such as morphine and codeine, semisynthetic molecules such as heroin and oxycodone, and synthetics such as pethidine and methadone. These drugs are powerful pain relievers, but with a high potential for dependence. Morphine administration (10 mg/kg/day for 6.5 days) increased VTA GFAP immunolabeling [151]. The opioid proinflammatory effect seems to last days after drug withdrawal. Indeed, morphine (12 mg/kg/day for 6 days) increased the protein levels of GFAP and p38 (a mitogen-activated protein kinase that regulates the release and production of inflammatory cytokines) in the NAc, 3 and 7 days after the last dose of morphine [152]. In this context, the administration of the anti-inflammatory ginger extract (100 mg/kg) before morphine injection reduced morphine-induced inflammatory markers in NAc [152]. Moreover, oxycodone (20 mg/kg/day) infused via osmotic minipumps for 12 days increased cytokines and growth factors (IL-1β, IL-2, IL-7, IL-9, IL-12, IL-15, IL17, M-CSF, VEGF) in PFC, 48 h after opiate withdrawal [153].

Tobacco-induced neuroinflammation: Nicotine is the main compound in tobacco and one of the most abused drugs in the world. A human study showed in smokers neuroinflammatory processes in NAc during nicotine withdrawal [154]. Exposure to tobacco for 4 weeks in rats increased gene expression of NF-κB and TNF-α in PFC, NAc, and VTA [155]. In addition, ceftriaxone treatment (a third-generation cephalosporin antibiotic) reversed the expression of NF-κB and TNF-α [155].

In summary, the previous evidence shows us that exposure to stimuli such as hypercaloric diets and drugs of abuse induces inflammatory processes in the mesocorticolimbic circuit, possibly participating in the maintenance of addictive behaviors. In this context, reducing the inflammatory condition in these brain areas may be a new therapeutic approach to treat serious conditions such as obesity and drug addiction.

### 3.2. Inflammaging, Gut Dysbiosis, and Highly Prevalent Diseases among Older Adults

The gut microbiota has a crucial role in maintaining the balance between pro- and anti-inflammatory responses [156,157]. The microbiota establishes a symbiotic relationship with the host, where individual environmental stimuli and genetic factors affect its composition, while various aspects of the host physiology are in turn regulated by these microorganisms [158]. In healthy individuals, the intestinal microbiota is multiple and diverse, containing 10^14^ bacteria, viruses, fungi, protozoa, and archaea [159], including between 1100 and 2000 bacterial taxa [160]. A healthy and diverse microbiota has been associated with the modulation of different physiological processes, such as immune function, metabolic homeostasis, insulin sensitivity, the permeability of the intestinal mucosa, detoxification, and regulation of gene expression [158]. Perturbations in gut microbiota, known as gut dysbiosis, have been shown in old-age patients [159]. Indeed, after 65 years of age, the resilience of the intestinal microbiota is reduced, as shown by a higher vulnerability to lifestyle changes, antibiotic treatment, and diseases that induce a reduction in the diversity of microbiota species [157]. Moreover, metagenomic analysis of stool samples obtained from sedentary, hospitalized, and healthy, self-reliant older adults showed that high biodiversity of the gut microbiota negatively correlates with sedentariness. In comparison, healthy, self-sufficient older adults showed a broad representation of bacterial species, indicating that the degree of physical activity in the elderly also seems to impact the gut microbiota diversity [161]. This elderly-associated intestinal microbiota shows less capacity to counteract pathogenic microorganisms and their metabolites, inducing an inflammatory response [162]. Elderly individuals show impaired immunity mechanisms of the intestinal mucosa that increase intestinal passage of bacterial LPS into the blood, triggering a chronic low-grade inflammatory state called inflammaging [156,163].

One of the possible consequences of inflammaging is the loss of muscle mass, typically observed in the elderly [156]. As recently shown, mice deficient in IL-10, a chronic inflammation and fragility syndrome model, have altered muscle mitochondria ultrastructure and autophagosomes in the myofibers [164], indicating that a chronic inflammatory state alters mitochondrial turnover in skeletal muscle, leading to the development of sarcopenia. One inflammatory pathway underlying the development of sarcopenia is Nod-like receptor protein-3 (NLRP3) inflammasome [165,166,167], which is involved in the activation of the proinflammatory cytokines IL-1β and IL-18 [168]. A sepsis state causes less muscular atrophy in NLRP3 KO mice [165]. Moreover, 24-month-old NLRP3^-/-^ mice presented fewer signs of sarcopenia, as shown by improved muscle strength and endurance, higher glycolytic potential, and reduced muscle atrophy, than old control mice [166,167]. Interestingly, one of the NLRP3 inflammasome activators is bacterial LPS [168], while butyrate behaves as a NLRP3 inflammasome inhibitor [169,170,171]. Mechanisms that reduce gut microbiota diversity also reduce butyrate production in old age [172]. Gram-negative bacteria, containing LPS as part of their cell wall, are overrepresented in the gut microbiota of sarcopenic patients, suggesting that sarcopenia is associated with a pro-inflammatory metagenome [173]. Meanwhile, bacteria from the genera Lachnospira, Fusicantenibacter, Roseburia, Eubacterium, and Lachnoclostridium—associated with butyrate production and possible anabolic effects on muscle—are reduced in the gut microbiota of sarcopenic older adults [173]. Taking these data together, it is plausible to think that butyrate production in the healthy gut microbiota of old people could have an anti-sarcopenic effect through the inhibition of NLRP3 inflammasome in skeletal muscle.

Similarly to the factors described above in relation to sarcopenia, alterations in gut microbiota in old age are also associated with neuroinflammatory conditions, such as dementia [174]. Sarcopenia could be considered a risk factor for developing dementia, especially Alzheimer’s disease (AD) [175], so a link between gut dysbiosis in old age, sarcopenia development, and dementia onset may exist. Patients with dementia show alterations in beta diversity and changes in taxonomic composition in gut microbiota when compared to controls; patients also present increased gut permeability and systemic inflammation [176]. Another article found a correlation between gut microbiota-associated metabolites and dementia development. Authors found that an increment in fecal ammonia was associated with an increased risk of dementia among older adults, while a decrease in fecal lactic acid reduced the risk of developing this condition [177]. A combination of elevated fecal ammonia and lactic acid content in the stools could be indicative of dementia [177], showing that changes in gut microbiota populations could modify the production of metabolites that can mediate pathologic processes. Currently, inflammation is considered one important mechanism in the pathogenesis of dementia [178]. For example, neuroinflammation is prevalent in AD and related dementias, with microglia and astrocyte activation around amyloid beta (Aβ) plaques and elevated levels of IL-1, IL-6, and TNF-β in the brain or blood [179,180,181]. Known conditions that produce an inflammatory state can potentiate the risk of developing dementia, for example, patients with obesity have an elevated risk of dementia. Women with central obesity had a 39% greater risk of dementia than women without central obesity [182]. Even sedentariness, another condition associated with gut dysbiosis [183] and low-grade inflammation [184,185], is positively related to the development of dementia [183]. During the COVID-19 pandemics, where obesity rates increased [186,187], lockdowns made it difficult to have an active life and increased sedentariness behavior among people [188]. Additionally, the direct consequences of COVID-19 infections for cognitive functions [189] are probably going to cause an increase in dementia rates in older adults. In summary, the inflammaging process experienced by older adults could be associated with changes in the gut microbiota and involved in the pathogenesis of highly prevalent diseases in this age group, such as sarcopenia and dementia, diseases that are certain to show increases in their incidence during and after the COVID-19 pandemic.

### 3.3. Neuroinflammation as a Common Feature of Neurological and Neurodegenerative Diseases

#### 3.3.1. Neuroinflammation as a Pathological Mechanism for Cognitive Impairment in Alzheimer’s Disease

Alzheimer’s disease (AD) is the most common neurodegenerative disease. According to WHO, AD was the seventh leading cause of death worldwide in 2019, and the second in high-income countries [190]. Cognitive impairment is considered a hallmark of AD, in which dysfunction in episodic memory, working memory, and executive function are the initial disturbances. As the condition develops, cognitive impairment becomes more intense and widespread. In the late stages of the disease, patients display profound behavioral change and impaired mobility, together with hallucinations and seizures, which strongly compromise their independence and quality of life [191,192]. At the neurophysiological level, AD patients display aberrant cortical oscillatory patterns and reduced large-scale functional connectivity, which correlates with cognitive performance [193,194,195]. Animal models of AD also display aberrant cortical and hippocampal oscillatory activity, with an impaired hippocampal encoding of space, and reduced synaptic plasticity [196,197,198]. This evidence suggests that neurophysiological processes involved in the implementation of cognitive functions are strongly disturbed in AD.

The classical “amyloid cascade hypothesis” for neurodegenerative diseases proposes that deposition of plaque β-amyloid (Ab) is the causative agent of AD pathology [199]. However, autopsies and imaging studies showed that amyloid deposition can be found in cognitively normal elderly subjects [200,201]. Moreover, it has been shown that the removal of Ab, both in animal and human models, did not modify the cognitive impairment in AD [202,203,204], suggesting an alternative pathological process for the cognitive impairment observed in AD.

Neuroinflammation has been proposed as a pathological mechanism involved in cognitive disturbance observed in AD [205,206,207]. Early postmortem studies revealed reactive microglia in different brain areas in AD [208]. Additionally, astrogliosis has been recognized in postmortem brain tissue from a subject with AD, and a correlation has been found between the degree of this alteration and cognitive impairment [205]. Interestingly, a significant correlation was found between cerebral makers of neuroinflammation in vivo, large-scale functional connectivity, and cognitive performance in AD patients [195]. Further, proinflammatory cytokines, including IL-1β, IL-6, TNF-α, and IL-8, were increased in serum and brain tissue of AD patients compared to controls [209,210]. Importantly, the reversion of neuroinflammation in animal models of AD recovered impaired neurophysiological and cognitive functions such as synaptic plasticity and spatial memory [211]. This evidence supports the role of neuroinflammation in the cognitive pathogenesis observed in AD [212].

Neuroinflammation in AD might be related to Ab accumulation, as postmortem analysis of the brains of AD patients showed reactive microglia colocalized with amyloid plaques [213,214]. Indeed, several amyloid peptides, fibrils, and amyloid precursor protein (APP) are potent glial activators, triggering an inflammatory response and microglial release of neurotoxic cytokines [215]. In the same line of evidence, the C-terminal 100 amino acid of APP, which is located in senile plaques, can induce astrogliosis [210]. This suggests a direct role of amyloid plaques in neuroinflammation. However, as mentioned before, Ab accumulation per se is not related to cognitive decline, which suggests that individual differences in inflammatory response could be related to the adverse effect of Ab accumulation. Recent genome-wide association studies (GWAS) have identified an interesting relationship between components of the innate immune system and the incidence of sporadic AD, supporting a link between the immune system (i.e., neuroinflammation) and the pathophysiology of dementia [216]. Interestingly, KO mice lacking the triggering receptor expressed on myeloid cells 2 (TREM2) gene, a risk factor for AD, which protein product regulates the release of cytokines [217], showed an increase in neuronal and synaptic loss accompanied by cognitive impairment [218]. Furthermore, upregulation of TREM2 reduced neuroinflammatory and cognitive defects observed in a transgenic rodent model of AD [219]. Additionally, monoclonal antibody binding membrane-associated and soluble TREM2 ameliorated cognitive dysfunction by inducing microglial activation and attenuating chronic neuroinflammation in AD mice [220]. Interestingly, considerable evidence suggests that polymorphisms of cytokines and other inflammatory genes seem to be a genetic risk factor for the development of AD [221]. Altogether, the evidence strongly suggests a pivotal role of neuroinflammation in the cognitive pathology observed in neurodegenerative diseases such as AD.

#### 3.3.2. Neuroinflammation as a Driver of Epilepsy

Epilepsy represents one of the most common brain diseases, affecting about 50 million people worldwide [222]. Epilepsy is characterized by the recurrent and unpredictable occurrence of seizures that may be or not be convulsive. During epileptogenesis, a progressive increase in neuronal excitability and hypersynchronization of electrical activity heightens the seizure susceptibility enough to produce the long-lasting disruption of brain functions known as status epilepticus [223]. The epileptic hyperexcitability is determined by an excitatory/inhibitory imbalance, which is attributable not only to the increase in glutamatergic neurotransmission but also to control loss of the network excitability via feedforward GABAergic inhibition [224,225,226,227]. In support of this idea, the evidence indicates that the extracellular glutamate is elevated up to 30 times in the epileptogenic human hippocampus during the seizure state, and the numbers of GABAergic neurons and synapses, as well as expression and function of GABA receptors, are strongly reduced [228,229,230,231]. However, this neurocentric vision about the cellular origin of epilepsy has started to change, and several pieces of evidence indicate that glial cells—microglia and astrocytes—contribute to both epileptogenesis and ictogenesis (seizure generation) [232].

Independently of the etiology, there is consensus that inflammation represents a common factor in most epilepsies and chronic epilepsy experimental models [233,234,235,236]. For example, after a variable period from the initial brain insult, recurrent seizures are followed by microglia activation. Activated microglia release several molecules depending on the stage of epileptogenesis, including inflammatory mediators (e.g., cytokines, chemokines, prostaglandins) and gliotransmitters (i.e., ATP, glutamate) [237]. In rodent and human epileptic tissue, IL-1β is released during the acute phase by activated microglia and astrocytes and only persists during epileptogenesis, chronically secreted by microglia, whereas IL-1 receptor 1 (IL-1RI) receptors are expressed exclusively by neurons and astrocytes starting from the acute phase [238]. Thus, these findings suggest that brain inflammation initiated by activated microglia represents a key factor in the onset and progression of this disease [233,237,239].

Although little is known about the direct mechanism by which inflammatory factors released from microglia exacerbate seizures and accelerate epileptogenesis, recent evidence suggests that gliotransmitters released by activated microglia may promote neuronal hyperexcitability. It has been shown that the pharmacological or genetic inactivation of the IL-1RI/TLR4 pathways diminishes neuronal hyperexcitability during epileptic seizures in chronic epilepsy models, as well as in pharmacoresistant epilepsy in humans [233,237,239,240]. Signaling via these receptors also acts through the activation of the purinergic receptor P2X7 subtype, an ATP-gated non-selective cation channel, which promotes Ca^2+^-dependent NLRP3 inflammasome activation, resulting in activation of caspase 1, and in turn, the production and release of IL-1β and IL-18 [241,242]. Thus, these inflammatory messengers, together with other immunological pathways chronically activated during epileptogenesis, have been associated with neurodegeneration and blood–brain barrier breakdown, which may contribute to the pathogenesis and maintenance of epilepsy [243,244]. Like microglia, reactive astrocytes change the expression pattern of several proteins, including transporters, hemichannels, receptors, and enzymes that are part of astrocyte–neuron and astrocyte–astrocyte signaling [245,246,247]. The astrocyte-to-neuron crosstalk—known as tripartite synapsis—forms a functional unit with neurons that directly regulates neuronal excitability and synaptic plasticity [248,249,250]. In purified cultures of microglia activated with LPS, as well as under neuroinflammatory conditions, cytokines along with ATP are released and activate the G-protein-coupled and ionotropic purinergic receptors—including P2Y1R and P2X expressed in the astrocytes [251,252,253,254]. Thus, in epileptic tissue, astrocytes respond to these molecules with an increase in astroglial Ca^2+^ elevation, upregulating the glutamate exocytosis from astrocytes, which increases the basal excitatory neurotransmission and exacerbates the seizure activity [244,255,256,257,258]).

In addition, IL-1α, TNF and C1q cytokines secreted by activated microglia are necessary and sufficient to induce the emergence of a subtype of neurotoxic astrocytes highly harmful to functional synapses [259]. TNFα also triggered the glutamate release from astrocytes of the inflamed hippocampus in an acute model of temporal lobe epilepsy (TLE), which required autocrine activation of P2Y1 receptors [253]. Several pieces of evidence suggest that ATP-mediated astrocyte–astrocyte signaling actively participates in the epileptiform activity. Astroglial Ca^2+^-mediated hyperexcitability in the epileptic hippocampus requires P2Y1R and pannexin-1 hemichannel activation in astrocytes, whereas the P2YR inhibitors and pannexin-1 blockade restore the normal astroglial Ca^2+^ pattern from hippocampus-kindled seizures [257,260,261]. Thus, abnormal glutamate released from reactive astrocytes increases synaptic efficacy by regulating the release probability at the glutamatergic terminal via the activation of group I metabotropic glutamate receptors (mGluR1/5), thereby setting the threshold for the induction of long-term plasticity at nearby glutamatergic synapses [249,250,262]. The upregulation of astrocyte–neuron signaling in epileptic tissue leads to spontaneous glutamate release from astrocytes, at the postsynaptic level, producing slow inward currents by extra-synaptic GluN2B-containing NMDA receptors on neighboring neurons, which can provide a source for dendritic plateau potentials that would generate simultaneous discharges in a neuronal population and promote synchronic activity at the neural network [263,264]. Thus, astrocytes can recruit and synchronize neurons for seizure activity, reducing the ictal threshold and promoting seizure propagation, which may result from the development of excitatory feedback loops between astrocytes and neurons [255,265,266].

In conclusion, ATP released from microglia activated by inflammatory factors can trigger the Ca^2+^-mediated activation of astrocytes, amplifying the astrocyte-to-astrocyte signaling and, in turn, upregulating the neuronal transmission and excitability by gliotransmitters (i.e., ATP and glutamate). However, whether the inflammatory process plays a causal role in epileptogenesis or forms part of the changes to transform a healthy brain into an epileptic one will require more novel and in-depth studies.

#### 3.3.3. Neuroinflammation Is a Common Mechanism for Different Factors in Exacerbated Anxiety

Anxiety is a natural reaction to stress in which an individual becomes alert and prepares to confront a stressor. On the other hand, anxiety disorder is defined as excessive anxiety, causing disproportionate fear and affecting everyday life. Common signs and symptoms of anxiety disorders are extreme anticipation and preoccupation with future concerns, which are normally accompanied by muscle tension, rapid heartbeats, and avoidance behavior for the stressor that causes the symptoms [267]. The prevalence of anxiety disorders is high, affecting an average of 20% of adults worldwide [268]. Moreover, elevated anxiety is also a prominent symptom of other mental disorders such as major depression, stress disorder, substance abuse [269,270], and neurological diseases such as Alzheimer’s disease [271] and epilepsy [272], suggesting common mechanisms involved in their generation.

Anxiety disorder is highly prevalent in people with inflammatory conditions such as rheumatoid arthritis [273], which suggests that inflammatory processes can be a pathological mechanism triggering increased anxiety. A comprehensive systematic review and meta-analysis study showed a positive correlation between levels of proinflammatory cytokines and the diagnosis of a generalized anxiety disorder (GAD) [274]; in this study, the authors analyzed 14 of 1718 identified studies, where the diagnosis of GAD and the measurement of cytokines from blood samples were matched, finding that elevated serum levels of C reactive protein (CRP) correlated with GAD [274]. Other circulating cytokines such as TNF-α, IL-6, and IFN-γ were also found to be elevated in a minority group of the 14 studies analyzed [274], suggesting that a generalized inflammatory state can be a biomarker of GAD, although further studies are still required.

Studies conducted with animal models show that a direct induction of inflammation in the brain nuclei responsible for emotional processing (the limbic circuitry) can induce increased anxiety. Indeed, an acute LPS systemic injection to mice induced microglia activation and increased the expression of NFκβ and the proinflammatory cytokines IL-1β and TNF-α in the basolateral amygdala (BLA); in addition, mice acutely treated with LPS showed increased anxiety behavior, as depicted by less time spent in exposed areas of the open-field test and the elevated plus maze test [275]. Importantly, these effects were accompanied by an increased excitability of projecting pyramidal neurons of the BLA and a disbalance to excitation as indicated by increased presynaptic glutamate release from mPFC afferents [275], suggesting that neuroinflammation caused by LPS-triggered microglia activation induces synaptic modifications underlying enhanced anxiety behavior. On the other hand, mice that chronically consumed ethanol for 5 months and then were deprived for 24 h showed increased anxiety, as depicted by decreased time in exposed areas of the elevated plus maze and dark and light box tests, and this effect was correlated with enhanced expression of the cytokines IL-1β and IL-17 in the striatum [276]. Remarkably, mice deficient in TLR2 or 4 with reduced cytokine release did not show anxiogenic behavior during alcohol abstinence [276], reinforcing the idea that increased inflammation mediated by cytokines in the brain induces anxiety behavior and revealing a crucial role for TLRs in the induction of anxiety.

The neuroinflammatory mechanisms induced by the nutritional state and diet can also contribute to the generation of increased anxiety. Thus, chronic high-fat diet consumption promoted anxiety behavior together with an increased level of the inflammatory cytokines IL-1β, IL-6, and TNF-α in the hippocampus [277]. The causal role of TNF-α generating anxiety behavior due to bad nutritional state was proven in a genetic mouse model of obesity, db/db mice. These mice show increased anxiety behavior and elevated cytokine levels in the hippocampus; importantly, the intra-hippocampal administration of etanercept, a TNF-α blocker, was sufficient to decrease anxiety behavior [278], further suggesting a pivotal role for neuroinflammation in the hippocampus as a key mechanism to induce anxiety. Other detrimental dietary patterns can also induce anxiety. Recently, it was shown that chronic consumption of a diet enriched in refined carbohydrates (HC) induces anxiety in mice [279], as with the case of HFD and obesity; mice chronically fed with HC had increased cytokine levels in the hippocampus and also increased nitrites in the PFC and hippocampus, an effect due to the overexpression of inducible nitric oxide synthase (iNOS) by microglia, leading to overproduction of NO [279]. Remarkably, the administration of aminoguanidine, an inhibitor of iNOS, reversed anxiety behavior induced by chronic HC diet, indicating that overproduction of NO was needed for the generation of anxiety behavior and illuminating iNOS as a potential target to treat anxiety disorders.

Together, the data indicate that increased anxiety can develop because of different inflammatory processes induced in brain nuclei of the limbic circuitry such as the BLA, hippocampus, and PFC. A wide range of factors can trigger this neuroinflammatory process, ranging from infection and peripheric inflammation to detrimental diet and nutritional state, positioning anxiety as a central common sign of different pathologies.

### 3.4. Peripheric Inflammation Contributing to the Generation of Cardiovascular Alterations

Cardiovascular diseases are responsible for 32% of all global deaths [280]. Among the risk factors associated with cardiovascular diseases, smoking, sedentarism, dyslipidemia, diabetes mellitus, arterial hypertension, and obesity are included [281]. It has been observed that clinically stable patients with prior acute events and symptomatic heart failure display evidence of chronic inflammation [282]. Moreover, this inflammation profile could aggravate cardiac dysfunction or predispose the patient to further decompensation [282]. The degree to which atherosclerotic plaques are formed, progress, and rupture depend, in part, on the degree of the inflammation in the plaque, and the expression of IL-1α and IL-1β is correlated with the progression of the atherosclerotic plaques, whereas minimal expression is found in healthy coronary arteries [282]. In addition, an increase in NLRP3 protein has been shown in the pericardium from patients with pericarditis, compared to healthy controls [283]. The NLRP3 inflammasome is an intracellular multiprotein complex that promotes the maturation of inflammatory cytokines, such as IL-1β and IL-18, that are rapidly induced in cardiovascular diseases upon infection, trauma, or stress [284]. In consequence, the NLRP3 inflammasome could mediate inflammation development as cardiovascular diseases progress [284].

In the heart, the priming phase of NLRP3 activation is mediated by several types of damage-associated molecular patterns (DAMPs). In chronic diseases such as atherosclerosis, arterial hypertension, obesity, and diabetes, metabolites and/or neurohormonal activation (angiotensin II, fatty acid, and glucose) promote high expression levels of NLRP3 inflammasome components that could be directly correlated with heart dysfunction and the rate of hospitalization in patients with dilated cardiomyopathy [285]. In this context, biopsies of these patients show increased cardiomyocyte pyroptosis, a programmed cell death induced by NLRP3 inflammasome [285,286,287,288].

Interestingly, at clinical levels, it has been observed that women have a lower risk of developing cardiovascular problems during their reproductive life (where the estrogen levels are high) compared to same-age men [283,289]. However, this advantage is lost when women start menopause (where the estrogen levels are low) [289,290]. Premenopausal women have decreased risk for cardiovascular disease compared to age-matched men, and decreased incidence of left ventricle hypertrophy and cardiac remodeling following myocardial infarction. On the other hand, postmenopausal women have an increased prevalence of cardiovascular diseases, reduced ischemic tolerance, and increased mortality following myocardial infarction compared to age-matched men [283,289]; this phenomenon also occurs in premenopausal women who have undergone ovariectomy, with the decrease or absence of estradiol responsible for this outcome (reviewed in [289]). In that context, it was shown that 17β-estradiol increases heat shock protein (HSP)70 expression in cardiac myocytes isolated from male and female rats in a dose-dependent manner, where the concentration necessary for female cardiomyocytes is higher compared to those of males [290,291]. The induction of HSP70 expression is through the activation of NFκB and heat shock factor (HSF)1 and can protect cardiomyocytes against damage induced by hypoxia [290]. In addition, another mechanism associated with the cardioprotection exerted by estradiol is antioxidant regulation [289,292], where a reduction in estradiol can induce an increase in mitochondrial oxidative stress [292], which, in turn, can activate the NLRP3 inflammasome [293,294]. Altogether, this body of evidence strongly suggests that cardiovascular diseases are a result of an inflammation process, and the cardioprotection that is seen in women within reproductive age compared to men is induced by estradiol, reducing the activation of the NLRP3 inflammasome and its mediators.

## 4. Clinical Perspectives

Inflammation is both a protective and a deleterious process, depending on the temporal course of bodily exposure to noxious stimuli and the grade of resolution of damage. Chronic mild-grade inflammation can originate in the periphery and the central nervous system and, in each division, induce local injury. In this review, we also discussed that peripheric inflammation (i.e., dysbiosis-induced gastrointestinal inflammation) harms the central level (i.e., neuroinflammation) and vice versa. Therefore, from a clinical point of view, we should see the organism as a whole unit where disruption of one tissue or system’s homeostasis can affect another. Remarkably, most factors that potentially trigger deleterious inflammation arise from the modern lifestyle (Figure 1). Thus, the first clinical approach should be prevention, pointing to promoting and establishing healthy habits from an early age. These habits should include frequent body exercise, frequent cognitive activities, prevention of chronic psychological stress, sleeping well, not using legally or illegally abused substances, and consuming a nutritive diet to promote microbiome diversity. Even if inflammation is already established, the change in lifestyle habits could significantly reduce inflammation [295].

The use of medications is another palliative alternative that can be complementary. Indeed, several anti-inflammatory drugs are currently being tested in different neuroinflammatory diseases [296]. On the other hand, other drugs are also being tested because of their anti-inflammatory potential, highlighting several from the family of antibiotics, plant derivatives, antagonists of chemokine receptors, etc. This strategy has the advantage of using already available drugs, which makes the pharmaceutical development process much faster. In the following table (Table 1), we list some of the drugs mentioned in this review that are currently being tested for their anti-inflammatory actions, the study in which they have been tested, the plausible biological pathway, and their therapeutical potential. In conclusion, the intricate interplay between inflammatory processes occurring in the CNS and peripheral tissues contributes significantly to the development and progression of various inflammatory diseases. This reciprocal relationship is particularly evident in the context of obesity, stress, neurodegenerative diseases, and metabolic disorders. The CNS can sense peripheral inflammation through intricate signaling mechanisms, triggering a cascade of neuroinflammatory responses that can further exacerbate peripheral inflammation. Conversely, peripheral inflammation can infiltrate the CNS via immune cells or soluble mediators, leading to neuroinflammation and neuronal dysfunction. Understanding and targeting these bidirectional interactions hold immense potential for therapeutic interventions aimed at ameliorating the impact of inflammatory diseases on both the CNS and peripheral tissues.

## Figures and Tables

**Figure 1 ijms-24-10083-f001:**
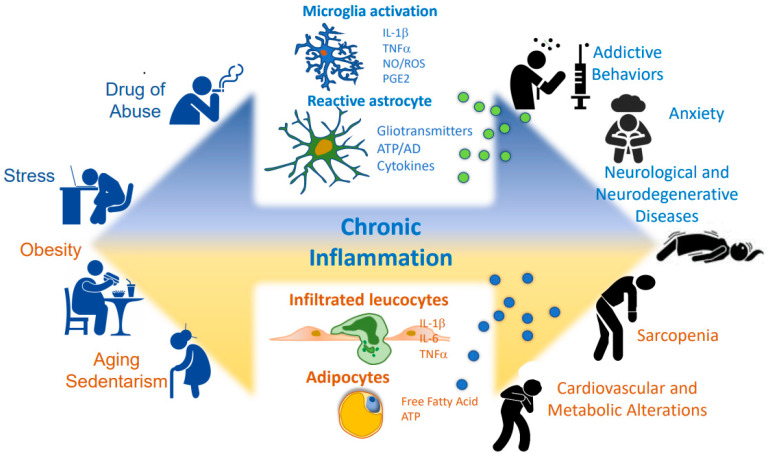
Role of Central and Peripheral Inflammation in Chronic Disease Development. Lifestyle factors such as drug consumption, stress, obesity, sedentary behavior, and conditions such as aging have been implicated in the development of chronic inflammation, a key driver of several chronic diseases. The underlying mechanisms involve the activation of inflammatory processes both in the CNS and in peripheral tissues. In the CNS, microglia and astrocytes are the key cell types that mediate the neuroinflammatory response, releasing pro-inflammatory cytokines and chemokines that contribute to neuronal damage and dysfunction. This can result in the development of neurodegenerative and neurological diseases, such as addictive behaviors. In the periphery, chronic inflammation could be induced by the infiltration of immune cells, such as macrophages and T cells, for example, into adipose tissue or skeletal muscle. This leads to the production of pro-inflammatory cytokines and chemokines, which can contribute to insulin resistance and the development of metabolic disorders such as type 2 diabetes and cardiovascular disease, or conditions such as sarcopenia, a pathology characterized by loss of muscle mass and function. These findings highlight the importance of lifestyle factors in inducing chronic inflammation and suggest that interventions aimed at modifying these factors may be effective in preventing or treating chronic inflammatory diseases.

**Table 1 ijms-24-10083-t001:** Drugs currently tested in preclinical or clinical studies for their anti-inflammatory actions as potential treatments in neuropsychiatric and neurodegenerative diseases.

Drug	Anti-Inflammatory Mechanism	Preliminary Findings	Therapeutic Potential	References
Nalmefene	Inhibition of ethanol-induced TLR4 activation	Reduction in ethanol consumption and the up-regulation of cytokines in PFC	Treatment for alcoholism	[139]
Ampicillin + sulbactam	Modulation of NMDA receptor NR2B subunits and HMGB1-associated pathways	Up-regulation of GLT-1 and mGluR5 in NAc shell, and reduction in HMGB1, RAGE and TNF-α	Treatment for alcoholism	[140]
Minocycline	Inhibition of microglial activation	Reduction in mRNA expression of IL-1β and IFN-γ microglial activation in the NAc; reduction in NAc protein levels of IBA-1, TNF-α, and IL-6 induced by METH	Treatment for obesity or drug addiction	[129,147]
SCH-23390	Antagonist of D1 receptors	Reduction in NAc protein levels of IBA-1, TNF-α and IL-6 induced by METH	Treatment for drug addiction	[147]
Box A	Antagonist of HMGB1	Reduction in protein levels of IL-1β in VTA, NAc and PFC in rats with acute METH administration	Treatment for drug addiction	[148]
Ginger extract	Not fully clear	Reduction in NAc morphine-induced inflammatory markers, p38 MAPK and GFAP	Treatment for drug addiction	[152]
Ceftriaxone	Not fully clear	Reduction in mRNA levels of NF-κB and TNF-α in PFC, NAc and VTA	Treatment for tobacco addiction	[155]
TREM 2 binding antibody	Monoclonal antibody binding the extracellular domain of human and murine TREM2	Improvement in cognitive function in experimental amyloidopathy models	Treatment for AD	[220]
A-438079 P2X7 receptor antagonist	Inhibition of microglial activation and purinergic-dependent inflammation.	Reduction in neuronal damage, and astroglial and microglial activation	Treatment for seizures and epilepsy	[242,297,298]
LPS-RS	TLR4 antagonist (VTA administration)	Reduction in cocaine-primed reinstatement	Treatment for drug addiction	[150]
Etanercept	TNF blocker	Decrease in anxiety behavior	Treatment for anxiety disorders	[278]
Aminoguanidine	iNOS blocker	Decrease in anxiety behavior	Treatment for anxiety caused by refined carbohydrates diet	[279]

## Data Availability

No new data were created for this manuscript.

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
