# Peer review of "Central and Peripheral Inflammation: A Common Factor Causing Addictive and Neurological Disorders and Aging-Related Pathologies"

_ijms, 2023, doi:10.3390/ijms241210083_

Round 1
Reviewer 1 Report
The review by Escobar et al made a good general overview regarding the role of inflammation in neurological and psychiatric disorders. They showed the evidences regarding the bidirectional pathway that links central nervous system and periphery. This is an interesting review that can help researchers to discuss about the theme.
My main question(s) is: do peripheral cells reach the central nervous system in health and/or disease? Do these cells play any role in the mentioned diseases (addiction, Alzheimer’s, epilepsy)? I think the authors should discuss more about it. Or are microglia and astrocytes the two only cellular components in the inflammatory orchestra involved in neurological disorders?
Minor:
Line 2: instead of “compromise the activation…” I would write “trigger the activation…”.
Between lines 54 and 56: the first two sentences of the paragraph need references/citation.
Lines 71 and 72: instead of “which is triggered by glial cells…” I would write “which is mediated by glial cells…”.
Document is well writen and English is fine.
Author Response
Dear Reviewer:
Thank you very much for your comments.
About this question: My main question(s) is: do peripheral cells reach the central nervous system in health and/or disease? Do these cells play any role in the mentioned diseases (addiction, Alzheimer’s, epilepsy)? I think the authors should discuss more about it. Or are microglia and astrocytes the two only cellular components in the inflammatory orchestra involved in neurological disorders?
We included discussion about that topic in lines 57-70.
Besides, we incorporated all the minors comments, Thank you very much.

Reviewer 2 Report
This review article by Escobar et al. discusses the links between peripheral and central inflammation, and additionally describes how inflammation can be promoted by various disease-associated risk factors. Inflammation is a very broad area of study because it is implicated in so many diseases, disorders and infections etc. This article was primarily focused on how brain inflammation might be related to peripheral inflammation in the context of chronic diseases.
The authors did a good job summarizing evidence that links different inflammatory processes in different systems and organs. Specifically, they made the case that inflammation is associated with diet, stress, addictive behavior, gut dysbiosis, Alzheimer’s Disease, Epilepsy, anxiety, and cardiovascular alterations. While the manuscript was well written and clear, to me it seemed like the paper sometimes jumped between ideas without clear connection. However, I find the article potentially acceptable for publication and only have minor comments that should be addressed.
Minor comments:
Line 56-57: It could be helpful to define astrocytes and microglia for the reader. i.e. what do they do under normal conditions?
Line 235-239: This is a very long, somewhat awkwardly worded sentence that could be revised for clarity.
Author Response
Dear Reviewer:
Thank you very much for your comments.
About this comment: Line 56-57: It could be helpful to define astrocytes and microglia for the reader. i.e. what do they do under normal conditions? We included discussion about that topic in lines 70-77.
and we corrected that sentence: Line 235-239: This is a very long, somewhat awkwardly worded sentence that could be revised for clarity.
Thank you very much
